# REM: Routing Entropy Minimization for Capsule Networks

## Abstract

Capsule Networks are biologically-inspired neural network models, but their interpretability still need to be further investigated. One of their main innovations relies on the routing mechanism which extracts a parse tree: its main purpose is to explicitly build relationships between capsules. However, their true potential has not surfaced yet: these relationships are extremely heterogeneous and difficult to understand, as the intra-class extracted parse trees are very different from each other. A school of thoughts, giving-up on this side, propose less interpretable versions of Capsule Networks without routing. This paper proposes REM, a technique which minimizes the entropy of the parse tree-like structure. We accomplish this by driving the model parameters distribution towards low entropy configurations, using a pruning mechanism as a proxy. Thanks to REM, we generate a significantly lower number of parse trees, with essentially no performance loss, showing also that Capsule Networks build stronger and more stable relationships between capsules.

## 1 Introduction

Capsule Networks (CapsNets) (Sabour et al., 2017; Hinton et al., 2018; Kosiorek et al., 2019) were recently introduced to overcome the shortcomings of Convolutional Neural Networks (CNNs). CNNs loose the spatial relationships between its parts because of max pooling layers, which progressively drop spatial information (Sabour et al., 2017). Furthermore, CNNs are also commonly known as "black-box" models: most of the techniques providing interpretation over the model are *post-hoc*: they produce localized maps that highlight important regions in the image for predicting objects (Selvaraju et al., 2017). CapsNets attempt to preserve and leverage an image representation as a hierarchy of parts, *carving-out* a parse tree from the networks. This is possible thanks to the iterative routing mechanism (Sabour et al., 2017) which models the connections between capsules. This can be seen as a parallel attention mechanism, where each active capsule can choose a capsule in the layer above to be its parent in the tree (Sabour et al., 2017). Therefore, CapsNets can produce interpretable representations encoded in the architecture itself (Sabour et al., 2017) yet can be still successfully applied to a number of applicative tasks (Zhao et al., 2019; Paoletti et al., 2018; Afshar et al., 2018).

However, understanding what really happens inside a CapsNet is still an open challenge. For a given input image, there are too many active co-coupled capsules, making the routing algorithm connections still difficult to understand, as the coupling coefficients typically have similar values, not exploiting the routing algorithm potential (Gu & Tresp, 2020). On the other hand, we would like for a given image to activate stronger and fewer connections between capsules, so that understanding and interpreting the parts-wholes relationships is a more straightforward process. To encourage this, we impose sparsity and entropy constraints. Furthermore, backward and forward passes of a CapsNet come at an enormous computational cost, since the number of trainable parameters is very high. For example, the CapsNet model deployed on the MNIST dataset by Sabour et al. (2017) is composed by an encoder and a decoder part. The full architecture has $8.2M$ of parameters. Do we really need such an amount of trainable parameters to achieve competitive results on such a task? Recently, many pruning methods were applied to CNNs in order to reduce the complexity of the networks, enforcing sparse topologies (Tartaglione et al., 2018; Molchanov et al., 2017; Louizos et al., 2018): is it possible to tailor one of these approaches with not only the purpose of lowering the parameters, but aiding the model's interpretability?

This work introduces REM (Routing Entropy Minimization) for CapsNets, which moves some steps towards the interpretability of the routing algorithm of CapsNets. Pruning can effectively reduce the overall entropy of the connections of the parse tree-like structure encoded in a CapsNet, because in low pruning regimes it removes noisy couplings which cause the entropy to increase considerably. We collect the coupling coefficients studying their frequency and cardinality, observing lower intra-class conditional entropy: the pruned version adds a missing explicit prior in the routing mechanism, grounding the coupling of the unused primary capsules disallowing fluctuations under the same baseline performance on the validation/test set. This implies that the parse trees are significantly less, hence more stable for the pruned models.

The rest of the paper is organized as follows: in Section 2 we introduce some of the basic concepts of CapsNets and their related works, in Section 3 we describe our technique called REM, in Section 4 we investigate the effectiveness of our method by testing it on many datasets and finally we discuss the conclusion of our work.

## 2 BACKGROUND AND RELATED WORK

This section first describes the fundamental aspects of CapsNets and their routing algorithm introduced by Sabour et al. (2017). Then, we review the literature especially related to sparsity in CapsNets.

**Capsule Networks Fundamentals.** CapsNets group neurons into *capsules*, namely activity vectors, where each capsule accounts for an object of one of its parts. Each element of these vectors accounts for different properties of the object such as its pose and other properties like color, deformation, etc. The magnitude of a capsule stands for the probability of existence of that object in the image. Typically, a CapsNet is composed by at least two capsule layers, called PrimaryCaps and DigitCaps (also called OutputCaps), with a total of $I$ and $J$ capsules respectively. The poses of $L$-th capsules $\boldsymbol{u_i}$, called *primary capsules*, are built upon convolutional layers. In order to compute the poses of the capsules of the next layer $L+1$, an iterative routing mechanism is performed. Each capsule $\boldsymbol{u_i}$ makes a prediction $\hat{\boldsymbol{u}}_{j|i}$, thanks to a transformation matrix $\boldsymbol{W}_{ij}$, for the pose of an upper layer capsule $j$

$$\hat{\boldsymbol{u}}_{j|i} = \boldsymbol{W}_{ij}\boldsymbol{u}_i. \tag{1}$$

Then, the total input $\boldsymbol{s}_j$ of capsule $j$ of the DigitCaps layer is computed as the weighted average of votes $\hat{\boldsymbol{u}}_{j|i}$

$$\boldsymbol{s}_j = \sum_i c_{ij}\hat{\boldsymbol{u}}_{j|i}, \tag{2}$$

where $c_{ij}$ are the coupling coefficients between a primary capsule $i$ and an output capsule $j$. The pose $\boldsymbol{v}_j$ of an output capsule $j$ is then defined as the normalized "squashed" $\boldsymbol{s}_j$

$$\boldsymbol{v}_j = \text{squash}(\boldsymbol{s}_j) = \frac{\|\boldsymbol{s}_j\|^2}{1 + \|\boldsymbol{s}_j\|^2} \frac{\boldsymbol{s}_j}{\|\boldsymbol{s}_j\|}. \tag{3}$$

So the routing algorithm computes the poses of output capsules and the connections between capsules of consecutive layers. The coupling coefficients are computed dynamically by the routing algorithm and they are dependent on the input. The coupling coefficients are determined by a "routing softmax" activation function, whose initial logits $b_{ij}$ are the log prior probabilities the $i$-th capsule should be coupled to the $j$-th one

$$c_{ij} = \text{softmax}(b_{ij}) = \frac{e^{b_{ij}}}{\sum_k .e^{b_{ik}}} \tag{4}$$

At the first step of the routing algorithm they are equals and then they are refined by measuring the agreement between the output $\boldsymbol{v}_j$ of the $j$-th capsule and the prediction $\hat{\boldsymbol{u}}_{j|i}$ for a given input. The agreement is defined as the scalar product $a_{ij} = \boldsymbol{v}_j \cdot \hat{\boldsymbol{u}}_{j|i}$. At each iteration, the update rule for the logits is

$$b_{ij} \leftarrow b_{ij} + a_{ij}. \tag{5}$$

The steps defined in equation 2, equation 3, equation 4, equation 5 are repeated for the $t$ iterations of the routing algorithm. The cross entropy loss is replaced with the *margin loss*.

**Capsule Networks Literature.** Capsule Networks were first introduced by Sabour et al. (2017) and since then a lot of work has been done, both to improve the routing mechanism and to build deeper models. Regarding the routing algorithm, Hinton et al. (2018) replace the dynamic routing with Expectation-Maximization, adopting matrix capsules instead of vector capsules. Wang & Liu (2018) model the routing strategy as an optimization problem. Li et al. (2018) use master and aide branches to reduce the complexity of the routing process. Peer et al. (2018) use inverse distances instead of the dot product to compute the agreements between capsules to increase their transparency and robustness against adversarial attacks. Hahn et al. (2019) incorporates a *self-routing* method such that capsule models do not require agreements anymore. De Sousa Ribeiro et al. (2020) replace the routing algorithm with variational inference of part-object connections in a probabilistic capsule network, leading to a significant speedup without sacrificing performance. Ribeiro et al. (2020) propose a new routing algorithm derived from Variational Bayes for fitting a mixture of transforming gaussians. Edraki et al. (2020) model entities through a group of capsule subspaces, without any form of routing. Since the CapsNet model introduced by Sabour et al. (2017) is a shallow network, several works attempted to build deep CapsNets. Rajasegaran et al. (2019) propose a deep capsule network architecture which uses a novel 3D convolution based dynamic routing algorithm aimed at improving the performance of CapsNets for more complex image datasets. Gugglberger et al. (2021) introduce residual connections to train deeper capsule networks.

**Sparse Capsule Networks.** A naive solution to reduce uncertainty within the routing algorithm is to simply run more iterations. As shown by Paik et al. (2019) and Gu & Tresp (2020), the routing algorithms tends to overly polarize the link strengths, namely a simple route in which each input capsule sends its output to only one capsule and all other routes are suppressed. On the one hand, this behavior is desirable because it makes the routing algorithm more interpretable, by making it possible to extract a parse tree thanks to this coupling coefficients. On the other hand, running many iterations is only useful in the case of networks with few parameters, as demonstrated by Renzulli et al. (2021), otherwise the performance will drop. Rawlinson et al. (2018) trained CapsNets in an unsupervised setting, showing that the routing algorithm does not discriminate among capsule anymore: the coupling coefficients collapse to the same value. Therefore, they sparsify latent capsule layers activities by masking output capsules according to a custom ranking function. Kosiorek et al. (2019) impose sparsity and entropy constraints into capsules, but they do not employ an iterative routing mechanism. Jeong et al. (2019) introduced a structured pruning layer called ladder capsule layers, which removes irrelevant capsules, namely capsules with low activities. Kakillioglu et al. (2020) solve the task of 3D object classification on point clouds with pruned Capsule Networks. Their objective was to compress robust capsule models in order to deploy them on resource-constrained devices.

The main contribution of our work relies on the fact that we regularize and prune the parameters in a CapsNet as a way to minimize the entropy of the connections computed by the routing algorithm. In fact, we show that relationships between objects and their parts in a standard CapsNets described by Sabour et al. (2017) have high entropies. We minimize these so that we can extract fewer parse trees. This allows us to effectively build dictionaries upon the input datasets and understand which are the shared object parts and transformations between different entities in the images.

## 3 ROUTING ENTROPY MINIMIZATION

The coupling coefficients computed by the routing mechanism model the part-whole relationships between capsules of two consecutive capsule layers. Assigning parts to objects (namely learning how each object is composed), is a challenging task. One of the main goals of the routing algorithm is to extract a parse tree of these relationships. Given the $\xi$-th input of class $j$, an ideal parse tree for a primary capsule $i$ detecting one of the parts of the entity in the input $\xi$ would ideally lead to

$$\boldsymbol{c}_{i-}^{\xi} = \mathbb{1}_{\hat{y}^{\xi}}, \tag{6}$$

where $\mathbb{1}_{\hat{y}^{\xi}}$ is the one-hot encoding for the target class $y^{\xi}$ of the $\xi$-th sample. This means that the routing process is able to carve a parse tree out of the CapsNet which explains perfectly the relationships between parts and wholes. One of the problems of this routing procedure is that there is no constraint on how many parse trees should be. In this section we present our technique REM, first showing how to extract a parse tree and then how to extract fewer parse trees. The pipeline of our method is depicted in Figure 1.

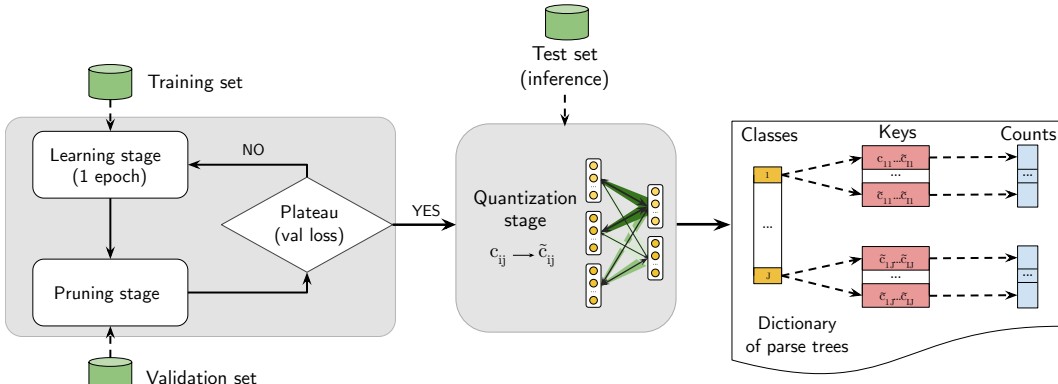

Figure 1: Pipeline of REM. After training is concluded, the coupling coefficients of the CapsNet are quantized, and the obtained parse trees are collected in a dictionary.

### 3.1 PARSE TREES EXTRACTION

Once we have a trained CapsNets model, in order to interpret the routing mechanism, we extract all the possible routing coupling coefficients and build a parse tree. Towards this end, we want to define a metric which helps us deciding if the relationships captured by the routing algorithm resemble a parse tree or not. Therefore, we organize the coupling coefficients into *associative arrays* so that we can compute the number of occurrences of each coupling sequence in order to measure the entropy of the whole dictionary. We refer to this entropy as the *simplicity* of the parse tree. In other words, we refer to the number of keys in the dictionary as the number of unique parse trees that can be carved-out from the input dataset. In the next paragraphs, we explain how to generate these sequences by discretizing the coupling coefficients and how to create the dictionary.

**Quantization.** During the quantization stage, we first compute the *continuous* coupling coefficients $c_{ij}^\xi$ for each $\xi$-th input example. It should be noticed that these are the coupling coefficients obtained after the forward pass of the last routing iteration. Then, we quantize them into $K$ discrete levels through the uniform quantizer $q_K(\cdot)$, obtaining

$$\tilde{c}_{ij}^\xi = q_K(c_{ij}^\xi). \tag{7}$$

We choose the lowest $K$ such that the accuracy is not deteriorated. We will here on refer to CapsNet+Q as trained CapsNet where the coupling coefficients are quantized.

**Parse tree extraction.** Given the quantized coupling coefficients of a CapsNet+Q, we can extract the parse tree (and create a dictionary of parse trees) for each class $j$, where each entry is a string composed by the quantization indices of the coupling coefficients. We will extract the coupling coefficients $\tilde{\boldsymbol{c}}_{-j}^\xi$ between the primary capsules $I$ and the predicted $j$-th output capsule. Given a dictionary for the coupling coefficients of a CapsNet+Q, we can compute the entropy for each class as

$$\mathbb{H}_j = -\sum_\xi \left\{ \mathbb{P}(\tilde{\boldsymbol{c}}_{-j}^\xi \mid y^\xi = j) \cdot \log_2 \left[ \mathbb{P}(\tilde{\boldsymbol{c}}_{-j}^\xi \mid y^\xi = j) \right] \right\} \tag{8}$$

where $\mathbb{P}(\tilde{\boldsymbol{c}}_{-j}^\xi \mid y^\xi = j)$ is the frequency of occurrences of a generic string $\xi$ for each *predicted* class $y^\xi$. Finally, the entropy of a dictionary for a CapsNet+Q on a given dataset is the average of the entropies $\mathbb{H}_j$ of each class

$$\mathbb{H} = \frac{1}{J} \sum_j \mathbb{H}_j. \tag{9}$$

Intuitively, the lower equation 9, the fewer the number of parse trees carved-out from the routing algorithm. We also target to obtain the distribution of these coupling coefficients. In general, we know that with $\Xi$ being the cardinality of the dataset, we have $\Xi \times I \times J$ coupling coefficients for the full dataset (with potential redundancies). Given the $i$-th primary capsule, however, we are only interested to $c_{ij}^\xi | y^\xi = j$. In this way, we reduce the coupling coefficients space to $I \times J$. We compute then the average of all the inputs belonging to an object class in order to output just $I \times J$ coupling coefficients.

## 3.2 Unconstrained routing entropy

In this subsection we are going to more-formally analyze the distribution of the coupling coefficients

$$c_{ij} = \frac{e^{b_{ij} + \sum_{r=1}^{t} \boldsymbol{v}_j^r \boldsymbol{u}_j \boldsymbol{W}_{ij}}}{\sum_k e^{b_{ik} + \sum_{r=1}^{t} \boldsymbol{v}_k^r \boldsymbol{u}_k \boldsymbol{W}_{ik}}} \tag{10}$$

where $t$ indicates the target routing iterations.[1] Let us evaluate the $c_{ij}$ over a non-yet trained model: as we saw also in Section 3.1, we have

$$c_{ij} \approx \frac{1}{J} \; \forall i, j. \tag{11}$$

When updating the parameters, following Gu & Tresp (2020), we have

$$\frac{\partial \mathcal{L}}{\partial \boldsymbol{W}_{ij}} = \left[ \frac{\partial \mathcal{L}}{\partial \boldsymbol{v}_j} \frac{\partial \boldsymbol{v}_j}{\partial \boldsymbol{s}_j} \cdot c_{ij} + \sum_{m=1}^{M} \left( \frac{\partial \mathcal{L}}{\partial \boldsymbol{v}_m} \frac{\partial \boldsymbol{v}_m}{\partial \boldsymbol{s}_m} \cdot \hat{\boldsymbol{u}}_{m|i} \frac{\partial c_{im}}{\hat{\boldsymbol{u}}_{m|i}} \right) \right] \cdot \boldsymbol{u}_i \tag{12}$$

where we can have the gradient for $\boldsymbol{W}_{ij} \approx 0$ in a potentially-high number of scenarios, despite $c_{ij} \neq \{0, 1\}$. Let us analyze the simple case in which we have perfect outputs, matching the ground truth, hence we are close to a local (or potentially the global) minimum of the loss function:

$$\left\| \frac{\partial \mathcal{L}}{\partial \boldsymbol{v}_m} \right\|_2 \approx 0 \; \forall m. \tag{13}$$

Looking at equation 4, we see that the right class is chosen, but given the squashing function, we have as an explicit constraint that, given the $j$-th class as the target one, we require

$$\|\boldsymbol{v}_j\|_2 \gg \|\boldsymbol{v}_m\|_2 \; \forall m \neq j \tag{14}$$

on the $\boldsymbol{W}_{ij}$, which can be accomplished in many ways, including:

- having sparse activation for the primary capsules $\boldsymbol{u}_i$: in this case, we have constant $\boldsymbol{W}_{ij}$ (typically associated to no-routing based approaches); however, we need heavier deep neural networks as they have to force sparse signals already at the output of the primary capsules. In this case, the coupling coefficients $c_{ij}$ are also constant by definition;
- having sparse votes $\hat{\boldsymbol{u}}_{j|i}$: this is a combination of having both primary capsules and weights $\boldsymbol{W}_{ij}$ enforcing sparsity in the votes, and the typical scenario with many routing iterations.

Having sparse votes, however, does not necessarily result in having sparse coupling coefficients: according to equation 5, the coupling coefficients are multiplied with the votes, obtaining the output capsules. The distribution of the coupling coefficients requires equation 14 to be satisfied only: if $\boldsymbol{W}_{ij}$ is not sparsely distributed, we can still have sparse votes. However, this is the main reason we observe high entropy in the coupling coefficient distributions: as the votes $\hat{\boldsymbol{u}}_{j|i}$ are implicitly sparse (yet also disordered, as we are not explicitly imposing any structure in the coupling coefficients distribution), the model is still able to learn but it finds a typical solution where $c_{ij}$ are not sparse. However, we would like to have sparsely distributed, recurrent couplings to the same $j$-th output caps $\boldsymbol{c}_{-j}$, establishing stable relationships between the features extracted at primary capsules layer.

Minimizing explicitly the entropy term equation 8 is an intractable problem due to the non-differentiability of the entropy term and of the quantization step (in our considered setup) and due to the huge computational complexity to be introduced at training time. Hence, we can try to implicitly enforce routing entropy minimization by forcing a sparse and organized structure in the coupling coefficients. Towards this end, one efficient solution is to enforce sparsity in the $\boldsymbol{W}_{ij}$ representation by compelling a vote between the $i$-th primary capsule and the $j$-th output caps to be exactly zero for any input, according to equation 10

$$c_{ij} = \frac{1}{\sum_k e^{b_{ik} + \sum_{r=1}^{t} \boldsymbol{v}_k^r \boldsymbol{u}_k \boldsymbol{W}_{ik}}}. \tag{15}$$

In this way, having a lower variability in the $c_{ij}$ values (and hence building more stable relationships between primary and output capsules), straightforwardly we are also explicitly minimizing the entropy of the quantized representations for the coupling coefficients. In the next subsection, we are going to tailor a sparsity technique to accomplish such a goal.

---

[1]for abuse of notation, in this subsection we suppress the index $\xi$

### 3.3 ENFORCING REM WITH PRUNING

CapsNets are trained via standard back-propagation learning, minimizing some loss function like margin loss. Our ultimate goal is to assess to what extent a variation of the value of some parameter $\theta$ would affect the error on the network output. In particular, the parameters not affecting the network output can be pushed to zero in a soft manner, meaning that we can apply an $L^2$ penalty term. A number of approaches have been proposed, especially in the recent years (Louizos et al., 2017; Molchanov et al., 2019; Lee et al., 2018). One recent state-of-the-art approach, LOBSTER (Tartaglione et al., 2022) proposes to penalize the parameters by their gradient-weighted $L^2$ norm, leading to the update rule

$$\theta^{t+1} = \theta^t - \eta G\left[\frac{\partial \mathcal{L}}{\partial \theta^t}\right] - \lambda \theta^t \text{ReLU}\left[1 - \left|\frac{\partial \mathcal{L}}{\partial \theta^t}\right|\right], \tag{16}$$

where $G\left[\frac{\partial \mathcal{L}}{\partial \theta^t}\right]$ is any gradient-based optimization update (for SGD it is the plain gradient, but other optimization strategies like Adam can be plugged) and $\eta, \lambda$ are two positive hyper-parameters.
Such a strategy is particularly effective on standard convolutional neural networks, and easy to plug in any back-propagation based learning system. Furthermore, LOBSTER is a regularization strategy which can be plugged at any learning stage, as it self-tunes the penalty introduced according to the learning phase: for this non-intrusiveness in the complex and delicate routing mechanism for CapsNets, it resulted in a fair choice to enforce REM.

## 4 EXPERIMENTS AND RESULTS

In this section we report the experiments and the results that we performed to test REM. We first show the results on the MNIST (Lecun et al., 1998) dataset, reporting also how the entropy and the accuracy values change during training. Then, we test REM on more complex datasets such as Fashion-MNIST (Xiao et al., 2017), CIFAR10 (Krizhevsky, 2009), SVHN (Netzer et al., 2011) and smallNORB (LeCun et al., 2004). We also performed experiments to test the robustness to affine transformations of CapsNets+REM. We used the same architectures configurations and augmentations described in Sabour et al. (2017).[2] We also conducted experiments applying our technique to $\gamma$-CapsNets (Peer et al., 2018), DeepCaps (Rajasegaran et al., 2019), Efficient-CapsNets (Mazzia et al., 2021) in order to test the efficacy of REM to some other variants of capsule models, including different architectures, routing algorithms and number of trainable parameters. We trained models with five random seeds. We report the classification accuracy (%) and entropy (averages and standard deviations), the *sparsity* (percentage of pruned parameters, median) and the number of keys in the dictionary (median). [3] The experiments were run on a NVIDIA Ampere A40 equipped with 48GB RAM, and the code uses PyTorch 1.10.

### 4.1 ABLATION STUDY

In order to assess our REM technique, we analyze in-depth the benefits of pruning towards REM on the MNIST dataset. Nowadays, despite its outdatedness, MNIST remains an omni-present benchmark for CapsNets (Sabour et al., 2017; Gu & Tresp, 2020; Rawlinson et al., 2018; Kosiorek et al., 2019; Keller & Welling, 2021).

**Entropy at different epochs**. On a given dataset, we target a model that has high generalization but low entropy, namely a low number of extracted parse trees. Figure 2 shows how the entropy (red line) and classification accuracy (black dotted line) changes as the sparsity increases during training. We can see that at the beginning of the training stage the entropy is low (1.83) because the routing algorithm has not learned yet to correctly discriminate the relationships between the capsules (97.83% of accuracy). This effect is almost the same when we train a CapsNet with $t = 1$ as Gu & Tresp (2020), where its entropy is exactly zero but capsules are uniformly coupled. However, at the end of the training process we can get a model trained with REM which has higher performances (99.60% of accuracy) and still low entropy (4.31).

---

[2]we have removed the decoder part of the network, see Appendix A.3.1 for more details.

[3]The code will be open-source released upon acceptance of the paper.

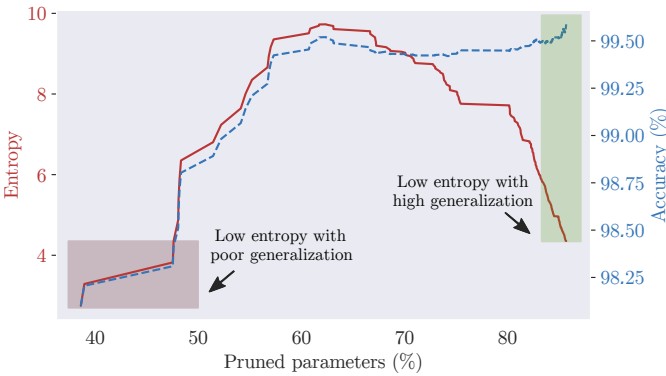

Figure 2: Accuracy and entropy curves vs pruned parameters on MNIST (test set).

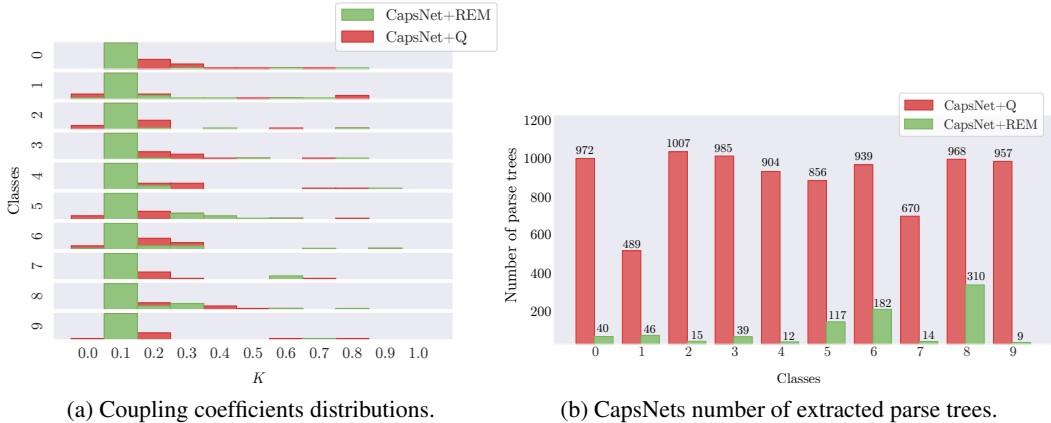

(a) Coupling coefficients distributions.   (b) CapsNets number of extracted parse trees.

Figure 3: Coupling coefficients distributions and number of parse trees for each class on MNIST (test set).

**Strength of parse trees**. In Figure 3a we plot the distributions of the coupling coefficients for a CapsNet+Q and a CapsNet+REM following the method described in Section 3.1. We can see that the distributions of the CapsNet+REM model are sparser that the ones for the CapsNet+Q model, namely we can carve-out parse trees with stronger part-whole relationships, achieving high generalization.

**Number of parse trees.** Figure 3b shows the number of intra-class parse trees (collected in a dictionary) for CapsNets+REM and a CapsNets+Q, namely a CapsNet where the quantization is applied without pruning the network during training. We can see that the number of keys of the dictionary for CapsNets+REM is lower than the one for CapsNets+Q for each class. Also the entropy measure for CapsNets+REM is lower compared to CapsNets+Q, namely, REM has successfully extract a lower number of parse trees on MNIST test set.

## 4.2 EXPERIMENTS

In this section we propose the experiments on more datasets. Considering the broad heterogeneity of proposed architectures, and the adaptability of REM also to other architectures, we have chosen to perform the experiments not only on CapsNets, but also to $\gamma$-CapsNets, DeepCaps and Efficient-CapsNets.

**Setup.** We trained and tested CapsNets on: i) Fashion-MNIST, 28×28 grayscale images (10 classes); ii) SVHN, 32×32 RGB images (10 classes); iii) smallNORB, 96×96 grayscale stereo images (5 classes) resized to 64×64 and cropped to 48x48 as Mazzia et al. (2021); iv) CIFAR10, 32×32 RGB images (10 classes).

| Model | Dataset | Accuracy | Sparsity | Entropy |
|-------|---------|----------|----------|---------|
| CapsNet+Q | Fashion-MNIST | $92.46\pm_{0.002}$ | 0 | $8.64\pm_{1.15}$ |
| CapsNet+REM | Fashion-MNIST | $92.62\pm_{0.001}$ | 80.71 | $4.80\pm_{1.70}$ |
| $\gamma$-CapsNet+Q | Fashion-MNIST | $92.43\pm_{0.01}$ | 0 | $3.98\pm_{0.76}$ |
| $\gamma$-CapsNet+REM | Fashion-MNIST | $93.01\pm_{0.01}$ | 87.07 | $1.45\pm_{0.68}$ |
| DeepCaps+Q | Fashion-MNIST | $92.33\pm_{0.002}$ | 0 | $7.15\pm_{1.33}$ |
| DeepCaps+REM | Fashion-MNIST | $94.61\pm_{0.0006}$ | 83.29 | $6.08\pm_{1.29}$ |
| Efficient-CapsNets+Q | Fashion-MNIST | $93.22\pm_{0.001}$ | 0 | $3.88\pm_{1.10}$ |
| Efficient-CapsNets+REM | Fashion-MNIST | $92.98\pm_{0.004}$ | 63.29 | $1.10\pm_{0.48}$ |
| CapsNet+Q | SVHN | $92.20\pm_{0.002}$ | 0 | $7.13\pm_{1.15}$ |
| CapsNet+REM | SVHN | $91.71\pm_{0.004}$ | 74.40 | $5.23\pm_{0.71}$ |
| $\gamma$-CapsNet+Q | SVHN | $87.42\pm_{0.12}$ | 0 | $7.15\pm_{0.86}$ |
| $\gamma$-CapsNet+REM | SVHN | $88.36\pm_{0.002}$ | 73.89 | $5.65\pm_{1.22}$ |
| DeepCaps+Q | SVHN | $93.20\pm_{0.003}$ | 0 | $11.06\pm_{0.58}$ |
| DeepCaps+REM | SVHN | $93.06\pm_{0.002}$ | 80.50 | $3.97\pm_{1.5}$ |
| Efficient-CapsNets+Q | SVHN | $93.62\pm_{0.0003}$ | 0 | $0.53\pm_{0.59}$ |
| Efficient-CapsNets+REM | SVHN | $93.12\pm_{0.0002}$ | 47.80 | $0.24\pm_{0.41}$ |
| CapsNet+Q | CIFAR10 | $78.42\pm_{0.026}$ | 0 | $6.26\pm_{0.61}$ |
| CapsNet+REM | CIFAR10 | $79.25\pm_{0.005}$ | 81.17 | $4.15\pm_{0.62}$ |
| $\gamma$-CapsNetQ | CIFAR10 | $73.08\pm_{0.004}$ | 0 | $3.67\pm_{0.70}$ |
| $\gamma$-CapsNet+REM | CIFAR10 | $74.89\pm_{0.002}$ | 90.22 | $3.22\pm_{0.66}$ |
| DeepCaps+Q | CIFAR10 | $90.47\pm_{0.001}$ | 0 | $8.99\pm_{0.52}$ |
| DeepCaps+REM | CIFAR10 | $90.35\pm_{0.001}$ | 46.83 | $7.07\pm_{1.01}$ |
| Efficient-CapsNets+Q | CIFAR10 | $81.51\pm_{0.005}$ | 0 | $0.25\pm_{0.35}$ |
| Efficient-CapsNets+REM | CIFAR10 | $81.49\pm_{0.004}$ | 53.79 | $0.005\pm_{0.02}$ |

Table 1: Accuracy (%), entropy and sparsity on Fashion-MNIST, SVHN and CIFAR10 (test set).

**Generalization ability.** As we can see in Tables 1, a CapsNet+REM has a high percentage of pruned parameters with a minimal performance loss. So this confirms our hypothesis that CapsNets are over-parametrized. We also report the entropy of the dictionary of the last routing layer for the quantized models. We can see that for all datasets when REM is applied to all models, even with fewer parameters that CapsNets such as Efficient-CapsNets, the entropy is successfully lower.

**Robustness to affine transformations.** To test the robustness to affine transformations of CapsNets+REM, we used expanded MNIST: a dataset composed by padded and translated MNIST, in which each example is an MNIST digit placed randomly on a black background of $40\times40$ pixels. We used the affNIST[4] dataset as test set, in which each example is an MNIST digit with a random small affine transformation. We tested an under-trained CapsNet with early stopping which achieved 99.22% accuracy on the expanded MNIST test set as in Sabour et al. (2017); Gu & Tresp (2020). We also trained these models until convergence. We can see in Table 2 that the under-trained networks entropies are high. Instead, a well-trained CapsNet+REM can be robust to affine transformations and have a low entropy.

**Robustness to novel viewpoints.** CapsNets are well known for their generalization ability to novel viewpoints (Sabour et al., 2017; Hinton et al., 2018). We conducted further experiments on smallNORB dataset to test the robustness to novel viewpoints of our technique following the experimental protocol of Hahn et al. (2019); Hinton et al. (2018) (more details can be found in the Appendix A.1.1). We employed Efficient-CapsNets, as they are the state-of-the-arts models on this dataset with a low number of trainable parameters. We used $K = 11$ quantization levels for Efficient-CapsNets+Q and Efficient-CapsNets+REM. In Table 3 we can see that Efficient-CapsNets+REM are indeed robust to novel viewpoints.

---

[4]https://www.cs.toronto.edu/ tijmen/affNIST/

| Model | expanded MNIST Accuracy (%) | affNIST Accuracy (%) | affNIST Sparsity (%) | affNIST Entropy |
|---|---|---|---|---|
| CapsNet+Q | 99.22 | $77.93\pm_{0.005}$ | 0 | $8.64\pm_{1.15}$ |
| CapsNet+REM | 99.22 | $81.81\pm_{0.008}$ | 71.26 | $8.45\pm_{1.10}$ |
| CapsNet+Q | $99.36\pm_{0.0005}$ | $83.14\pm_{0.002}$ | 0 | $8.45\pm_{0.99}$ |
| CapsNet+REM | $99.48\pm_{0.0002}$ | $85.23\pm_{0.001}$ | 87.32 | $5.93\pm_{1.39}$ |

Table 2: Results on affNIST test set for under-trained and well-trained models.

| Model | Familiar | | Novel | | Sparsity |
|---|---|---|---|---|---|
| | Accuracy | Entropy | Accuracy | Entropy | |
| Efficient-CapsNet+Q ($\phi$) | $89.79\pm_{0.08}$ | $2.03\pm_{0.28}$ | $78.25\pm_{0.011}$ | $2.38\pm_{0.17}$ | 0 |
| Efficient-CapsNet+REM ($\phi$) | $90.20\pm_{0.11}$ | $1.11\pm_{0.27}$ | $78.18\pm_{0.013}$ | $1.16\pm_{0.06}$ | 55.34 |
| Efficient-CapsNet+Q ($\psi$) | $89.22\pm_{0.09}$ | $1.94\pm_{0.93}$ | $79.52\pm_{0.010}$ | $1.88\pm_{1.03}$ | 0 |
| Efficient-CapsNet+REM ($\psi$) | $88.85\pm_{0.06}$ | $1.09\pm_{0.94}$ | $78.69\pm_{0.015}$ | $1.05\pm_{0.94}$ | 47.91 |

Table 3: Accuracy (%) and entropy values on the smallNORB test set on familiar and novel viewpoints (elevations $\phi$ and azimuths $\psi$) seen and unseen during training respectively.

**Improved interpretability with REM.** Since CapsNets are typically stacked on top of convolutional layers, capsules can also have a spatial connotation. Therefore, we use the coupling coefficients values as a visual attention built-in explanation to carve-out the part-structure discovered by a capsule model. We follow (Gu, 2021), where the coupling coefficients of the predicted class $j$ of a trained model for a given input is used as attention matrix. Unlike (Gu, 2021), we also weight each coupling coefficient $\tilde{c}_{ij}$ by the activation $\|\boldsymbol{u}_i\|$ of the corresponding primary capsule $i$. We upsampled the saliency map to the input size with the bilinear method. Figure 4 shows the saliency maps overlaid on some CIFAR10 images (for more details on how to extract the saliency map see Appendix A.1.5). We can see that the part-whole hierarchies extracted with REM are more succinct and human-interpretable. For example, in order to classify an object as an airplane, the network detect the wings and tail as discriminating parts. As regards the car it detects not only the road but also the window, the door and the wheels. Finally, as regards the horse, CapsNet+REM correctly detects its head, main and legs.

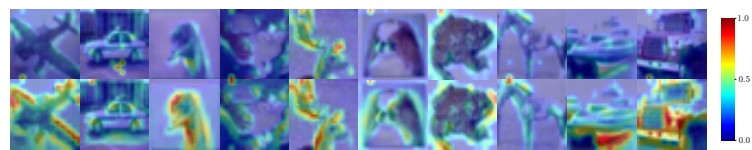

Figure 4: Saliency maps for CIFAR10 for CapsNet+Q (above) and CapsNet+REM (below).

## 5  CONCLUSION

This paper moved some steps towards an improved interpretability of the routing algorithm in CapsNets with REM (Routing Entropy Minimization), which drives the model parameters distribution towards low entropy configurations. We first showed how to extract the parse tree of a CapsNet by discretizing its connections and then collecting the possible parse trees in associative arrays. Standard CapsNets show high entropy in the parse trees structures, as an explicit prior on the coupling coefficients distribution is missing. Indeed, the number of intra-class generated parse trees is relatively high. We showed how pruning methods, in low pruning regimes, naturally reduce such entropy as well as the cardinality over the possible parse trees, testing such a phenomenon on several datasets. We also showed that REM can also carve-out parse trees with stronger part-whole relationships, achieving high generalization. Furthermore, we empirically observe that a CapsNet+REM model remains robust to affine transformations and novel viewpoints. REM opens research pathways towards the distillation of parse trees and model interpretability, including the design of a pruning technique specifically-designed for REM.

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

# A APPENDIX

## A.1 EXPERIMENTS DETAILS

In this section we provide the technical details of our experiments, including the datasets setup, the optimizers, hyperparameter values and architectures configurations.

### A.1.1 DATASETS SETUP

For MNIST, Fashion-MNIST and CIFAR10 we used 5% of the training set as validation set. To test the robustness to novel azimuths on smallNORB, we train all models on 1/3 of training data with azimuths of 0, 20, 40, 300, 320, 340 degrees and test them on 2/3 of test data with remaining azimuths never seen during training. In order to test the robustness of our technique on novel elevations, we trained models on 1/3 of training data with elevations of 30, 35, 40 degrees from the horizontal, and tested on 2/3 of test data with the remaining elevations. For Tiny Imagenet we used 10% of the training set as validation set and the original validation set as test set. Finally, to test the robustness to affine transformations, we used expanded MNIST training and validation sets (40×40 padded and translated MNIST images) and the affNIST test set, in which each example is an MNIST digit with a random small affine transformation.

### A.1.2 MODEL ARCHITECTURES

All models employed in this work were tested using the same architectures (number of layers, capsule dimensions, number of routing iterations etc.) presented in the original papers. Therefore, for CapsNets, $\gamma$-CapsNets, DeepCaps and Efficient-CapsNets we used the same architectures configurations as in Sabour et al. (2017); Peer et al. (2018); Rajasegaran et al. (2019); Mazzia et al. (2021) respectively.

### A.1.3 TRAINING

For CapsNets+Q, $\gamma$-CapsNet+Q, DeepCaps+Q and Efficient-CapsNets+Q we take the model that achieved the lowest loss on the validation set, while for CapsNets+REM, $\gamma$-CapsNet+REM, Deep-Caps+REM and Efficient-CapsNets+REM we take the model on the last epoch. We checked the loss on the validation set and we used an early-stop of 200 epochs. The models were trained on batches of size 128 using Adam optimizer with its PyTorch 1.10 default parameters, including an exponentially decaying learning rate factor of 0.99.

### A.1.4 CHOICE OF QUANTIZATION LEVELS

The routing algorithms used in the models employed in this paper are performed between two consecutive capsule layers. As we can see in Figure 5, the choice of the number of quantization levels $K$ for the coupling coefficients computed by a routing algorithm of a CapsNet affects the performance of the network. We select the value for $K$ that achieves the best accuracy value with relatively low entropy. In this case, when $K$=11, CapsNet+Q achieves 99.47% accuracy and 9.32 entropy, while CapsNet+REM achieves 99.57% accuracy and 4.40 entropy. When stacking multiple capsule layers, for example using $\gamma$-CapsNets and DeepCaps, we apply the quantization stage to each of this layers and we compute the entropy values on the last layer. For each capsule layer, we chose the lowest $K$ such that the accuracy is not deteriorated. For example, we used $\gamma$-CapsNets with 3 capsule layers as in Peer et al. (2018). For $\gamma$-CapsNets+Q and $\gamma$-CapsNets+REM we found $K = 11$ for the first two capsule layers and $K = 6$ for the last two. For DeepCaps+Q and DeepCaps+REM we used $K = 11$ for all the capsule layers where the number of routing iterations is greater than one. For Efficient-CapsNets+Q and Efficient-CapsNets+REM we used $K = 11$ on smallNORB.

As regards CapsNets+Q and CapsNets+REM, on MNIST, Fashion-MNIST, CIFAR10 and affNIST, we found $K = 11$ for the quantizer, while for Tiny ImageNet we found $K = 129$ quantization levels.

### A.1.5 HOW TO EXTRACT THE SALIENCY MAP

Figure 6 depict a visualization of our method to extract a saliency map from an input image given a CapsNet model. We build a saliency map, or explanation map, $\mathbf{E}^\xi$ for a given input $\xi$ exploiting

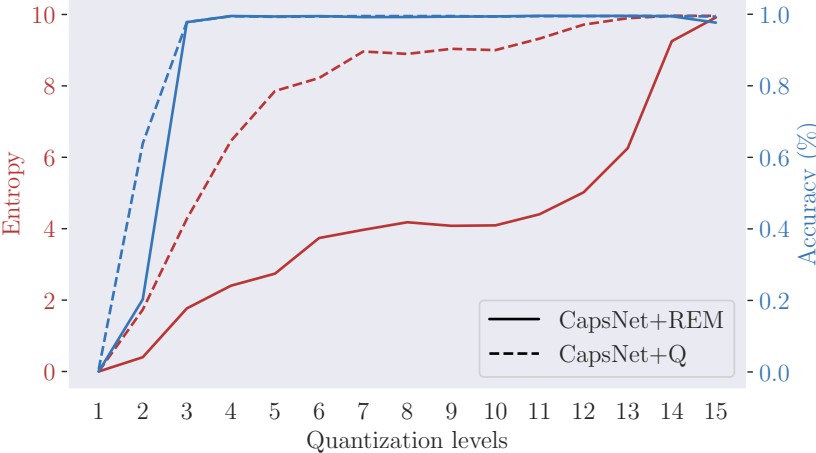

Figure 5: Entropy and accuracy values for CapsNet+Q and CapsNet+Q with different quantization levels on MNIST (test set).

the quantized coupling coefficients of the predicted class and the activations of the primary capsules. This allows us to carve-out the part-structure of the object in the image. Note that for simplicity of notation, in the previous Sections, we omitted the spatial dimensions of primary capsules. But since primary capsules are built upon a convolutional layer, we refer to $\boldsymbol{u}_{nm}^{\xi}$ to indicate the pose of the primary capsule in position $(n, m)$ for a given input $\xi$. With $\tilde{c}_{mnj}^{\xi}$, we refer to the quantized coupling coefficient between the primary capsule in position $(n, m)$ and the predicted class $j$ for a given input $\xi$. Therefore, each element of the explanation map is computed using

$$\mathbf{E}_{mn}^{\xi} = \|\boldsymbol{u}_{nm}^{\xi}\| * \tilde{c}_{mnj}^{\xi} \tag{17}$$

Then we upsampled the saliency map to the input size with the bilinear method.

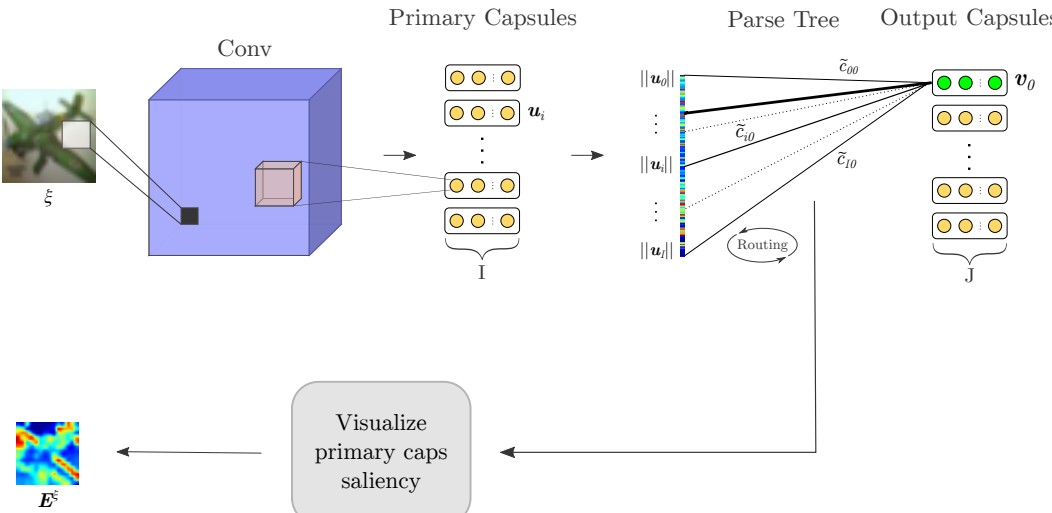

Figure 6: Extraction of the saliency map given an input image of label 0 (airplane) and a CapsNet model.

## A.2 ADDITIONAL AND EXTENDED RESULTS

In this section we provide additional and extended results for MNIST, Fashion-MNIST, SVHN, CI-FAR10, smallNORB, affNIST and Tiny ImageNet, including distributions of the coupling coefficients.

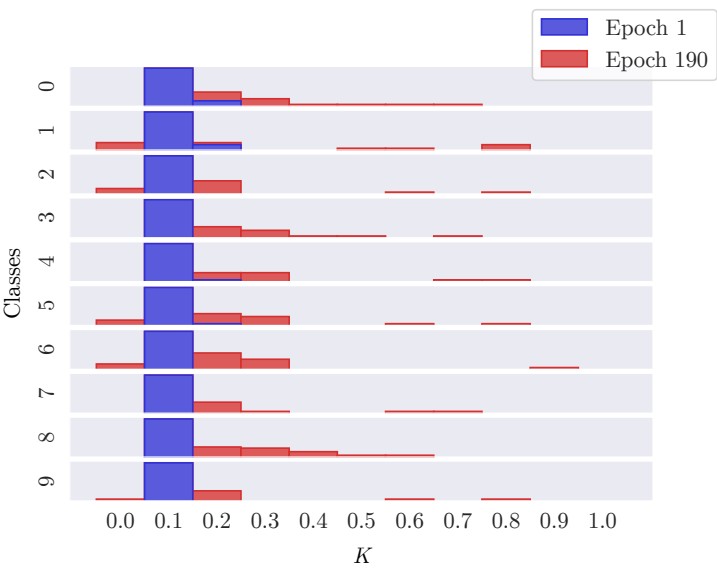

Figure 7: Coupling coefficients distributions for each class of two CapsNets+Q at epochs 1 and 190 on MNIST (test set).

We also provide additional visualizations employing the dictionary built with our technique REM in order to give a better understanding of what is the impact of having fewer parse trees with stronger connections.

### A.2.1 DISTRIBUTIONS AND TABLES

Figure 7 shows the distributions of the coupling coefficients for each class on MNIST of two CapsNets+Q at epochs 1 and 190. It can be observed that after the first epoch CapsNet is clearly far from optimality, both in term of performance (accuracy of 97.4%) and parse tree interpretability: indeed all coupling coefficients are almost equal to the value selected for initialization, i.e. $1/J$, where $J$ is the number of output capsules. Table 4 shows the performances on MNIST of $\gamma$-CapsNets, DeepCaps and Efficient-CapsNets. We can notice that $\gamma$-CapsNet and $\gamma$-CapsNet+REM has the lowest entropy values, since $\gamma$-CapsNets employ a scaled-distance-agreement routing algorithm which enforces the single parent constraint. With our technique REM we can successfully lower the entropy even more. Table 5 reports the accuracy of CapsNets without the quantization stage. Table 6 reports the accuracy, sparsity and entropy values for CapsNet+Q and CapsNet+REM on Tiny ImageNet. Table 7 reports the number of parse trees for Fashion-MNIST, SVHN, CIFAR10, affNIST and Tiny ImageNet (only the first ten classes). We conducted further experiments on smallNORB dataset to test the robustness to novel viewpoints of our technique on CapsNets, $\gamma$-CapsNets and DeepCaps. The results are shown in Table 10, which is an extended version of Table 3. We also show in Table 11 the performances of these networks without quantization. All the models are trained with our own implementations when the source code is not available. We can see that the number of parse trees and entropies for CapsNets+REM is lower than the one for CapsNets+Q, also for these datasets.

### A.3 FASHION-MNIST SALIENCY MAPS

Here we show in Figure 8 the saliency maps for Fashion-MNIST generated using the method described in Appendix A.1.5. We can notice that understanding which are the parts of an object that the model relied on to assign it the predicted label is more straightforward and human-interpretable in CapsNets+REM. For example, the network is able to recognize the sleeves and collar as distinguishing features of a t-shirt or a sweater, the cuffs and the cronch for trousers and the handles for the bag.

| Model | Parameters | Accuracy (%) | Sparsity (%) | Entropy |
|---|---|---|---|---|
| CapsNet+Q | 6.8M | $99.56\pm_{0.0003}$ | — | $9.53\pm_{0.54}$ |
| CapsNet+REM | 0.9M | $99.56\pm_{0.0002}$ | 85.53 | $4.16\pm_{1.59}$ |
| $\gamma$-CapsNet+Q | 7.7M | $99.50\pm_{0.0007}$ | — | $1.87\pm_{1.38}$ |
| $\gamma$-CapsNet+REM | 0.8M | $99.50\pm_{0.0005}$ | 89.71 | $1.34\pm_{1.09}$ |
| DeepCaps+Q | 8.4M | $99.51\pm_{0.0024}$ | — | $5.26\pm_{2.00}$ |
| DeepCaps+REM | 2.4M | $99.61\pm_{0.0023}$ | 71.73 | $3.10\pm_{1.07}$ |
| Efficient-CapsNets+Q | 161k | $99.55\pm_{0.003}$ | — | $4.38\pm_{1.59}$ |
| Efficient-CapsNets+REM | 43k | $99.58\pm_{0.006}$ | 73.15 | $2.60\pm_{1.72}$ |

Table 4: Results for CapsNets, $\gamma$-CapsNets, DeepCaps and Efficient-CapsNets on MNIST (test set).

| Model | Dataset | Accuracy | Sparsity |
|---|---|---|---|
| CapsNet | MNIST | $99.57\pm_{0.0002}$ | 0 |
| CapsNet | MNIST | $99.58\pm_{0.0003}$ | 85.53 |
| $\gamma$-CapsNet | MNIST | $99.53\pm_{0.0015}$ | 0 |
| $\gamma$-CapsNet | MNIST | $99.51\pm_{0.0009}$ | 89.71 |
| DeepCaps | MNIST | $99.58\pm_{0.0032}$ | 0 |
| DeepCaps | MNIST | $99.63\pm_{0.0019}$ | 71.73 |
| Efficient-CapsNets | MNIST | $99.57\pm_{0.004}$ | 0 |
| Efficient-CapsNets | MNIST | $99.61\pm_{0.003}$ | 73.15 |
| CapsNet | Fashion-MNIST | $92.76\pm_{0.002}$ | 0 |
| CapsNet | Fashion-MNIST | $93.09\pm_{0.001}$ | 80.71 |
| $\gamma$-CapsNet | Fashion-MNIST | $92.59\pm_{0.01}$ | 0 |
| $\gamma$-CapsNet | Fashion-MNIST | $93.47\pm_{0.002}$ | 87.07 |
| DeepCaps | Fashion-MNIST | $92.36\pm_{0.002}$ | 0 |
| DeepCaps | Fashion-MNIST | $94.63\pm_{0.0006}$ | 83.29 |
| Efficient-CapsNets | Fashion-MNIST | $93.31\pm_{0.002}$ | 0 |
| Efficient-CapsNets | Fashion-MNIST | $93.28\pm_{0.003}$ | 63.29 |
| CapsNet | SVHN | $93.30\pm_{0.002}$ | 0 |
| CapsNet | SVHN | $92.81\pm_{0.004}$ | 74.40 |
| $\gamma$-CapsNet | SVHN | $89.02\pm_{0.001}$ | 0 |
| $\gamma$-CapsNet | SVHN | $90.72\pm_{0.001}$ | 73.89 |
| DeepCaps | SVHN | $93.32\pm_{0.003}$ | 0 |
| DeepCaps | SVHN | $93.16\pm_{0.002}$ | 80.50 |
| Efficient-CapsNets | SVHN | $93.64\pm_{0.0004}$ | 0 |
| Efficient-CapsNets | SVHN | $93.14\pm_{0.0002}$ | 47.80 |
| CapsNet | CIFAR10 | $79.93\pm_{0.002}$ | 0 |
| CapsNet | CIFAR10 | $80.33\pm_{0.005}$ | 81.17 |
| $\gamma$-CapsNet | CIFAR10 | $74.02\pm_{0.002}$ | 0 |
| $\gamma$-CapsNet | CIFAR10 | $75.06\pm_{0.002}$ | 90.22 |
| DeepCaps | CIFAR10 | $90.80\pm_{0.001}$ | 0 |
| DeepCaps | CIFAR10 | $90.94\pm_{0.002}$ | 46.83 |
| Efficient-CapsNets | CIFAR10 | $81.53\pm_{0.005}$ | 0 |
| Efficient-CapsNets | CIFAR10 | $81.51\pm_{0.004}$ | 53.79 |
| CapsNet | Tiny ImageNet | $60.85\pm_{0.19}$ | 0 |
| CapsNet | Tiny ImageNet | $58.96\pm_{0.11}$ | 44.27 |

Table 5: Accuracy and sparsity results without quantization on MNIST, Fashion-MNIST, SVHN, CIFAR10 and Tiny ImageNet (test set).

| Model | Accuracy | Sparsity | Entropy |
|---|---|---|---|
| CapsNet+Q | $58.50\pm_{0.25}$ | 0 | $5.18\pm_{0.67}$ |
| CapsNet+REM | $54.02\pm_{0.14}$ | 44.27 | $3.15\pm_{0.81}$ |

Table 6: Accuracy (%), entropy and sparsity results on Tiny ImageNet (test set).

| Class | F-MNIST | | affNIST | | CIFAR-10 | | T-ImageNet | | SVHN | |
|---|---|---|---|---|---|---|---|---|---|---|
| | Q | REM | Q | REM | Q | REM | Q | REM | Q | REM |
| $\#_0$ | 610 | 58 | 9640 | 248 | 345 | 64 | 65 | 28 | 585 | 74 |
| $\#_1$ | 936 | 140 | 5490 | 66 | 407 | 125 | 57 | 22 | 1750 | 85 |
| $\#_2$ | 332 | 35 | 14055 | 438 | 375 | 115 | 50 | 9 | 1985 | 302 |
| $\#_3$ | 600 | 80 | 4446 | 97 | 200 | 70 | 12 | 11 | 1054 | 266 |
| $\#_4$ | 346 | 30 | 9059 | 161 | 291 | 39 | 39 | 15 | 890 | 60 |
| $\#_5$ | 828 | 35 | 7732 | 425 | 235 | 76 | 51 | 7 | 1116 | 129 |
| $\#_6$ | 297 | 23 | 11957 | 1244 | 282 | 58 | 90 | 14 | 780 | 70 |
| $\#_7$ | 915 | 31 | 3109 | 164 | 381 | 71 | 48 | 21 | 676 | 31 |
| $\#_8$ | 812 | 217 | 15521 | 396 | 305 | 37 | 129 | 30 | 434 | 139 |
| $\#_9$ | 978 | 615 | 3703 | 67 | 147 | 52 | 35 | 12 | 681 | 110 |

Table 7: Number of parse trees for each class of a CapsNet+Q and CapsNet+REM on Fashion-MNIST, affNIST, CIFAR-10, Tiny ImageNet and SVHN.

### A.3.1 DECODER

A CapsNet is typically composed of an encoder and a decoder part, where the latter is a reconstruction network with 3 fully connected layers Sabour et al. (2017). In the previously-discussed experiments, we have removed the decoder. One limitation of our work arises when computing the entropy of CapsNets trained with the decoder. Tables 8 and 9 reports the classification results and entropies values respectively when we trained the encoder and the decoder part together. We observed that the entropy of a CapsNets+REM is almost the same as that of a CapsNet+Q. Indeed, when the decoder is used, the activity vector of an output capsule encodes richer representations of the input. Sabour et al. (2017) introduced the decoder to boost the routing performance on MNIST by enforcing the pose encoding a capsule. They also show that, when a perturbed activity vector is fed to the decoder, such perturbation affects the reconstruction. So capsules representations are *approximately equivariant*, meaning that even if they do not come with guaranteed equivariances, transformations applied to the input can still be described by continuous changes in the output vector. In order to verify if output capsules of a trained CapsNet+REM without the decoder (so with low entropy) are still approximately equivariant, we stacked on top of it the reconstruction network, without training the encoder. The decoder on MNIST dataset is composed by 3 fully connected layers of 512, 1024 and 784 neurons respectively with two RELU and a final sigmoid activation functions. This network is trained minimizing the euclidean distance between the image and the output of the sigmoid layer. We can see in Figure 9 that CapsNets+REM with low entropy are still approximately equivariant to many transformations.

| Model | MNIST | F-MNIST | CIFAR10 |
|---|---|---|---|
| CapsNet+Q | $99.58\pm_{0.0003}$ | $92.57\pm_{0.003}$ | $72.40\pm_{0.005}$ |
| CapsNet+REM | $99.63\pm_{0.0002}$ | $92.76\pm_{0.003}$ | $76.00\pm_{0.006}$ |

Table 8: Classification results with the decoder on MNIST, Fashion-MNIST, CIFAR10 (test set).

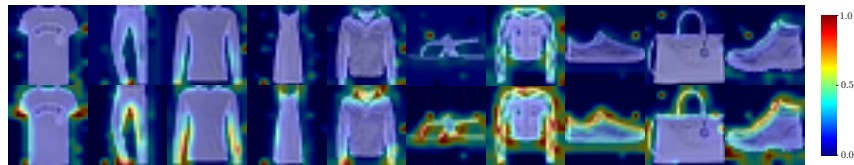

Figure 8: Saliency maps for Fashion-MNIST for CapsNet+Q (above) and CapsNet+REM (below).

Width + translation 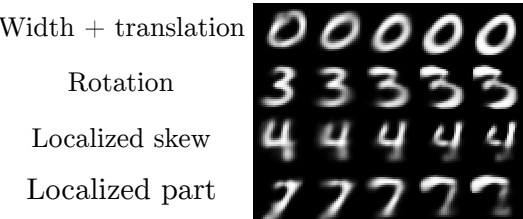
Rotation
Localized skew
Localized part

Figure 9: MNIST perturbation reconstructions of a freezed CapsNet+REM.

| Model | MNIST | F-MNIST | CIFAR10 |
|-------|-------|---------|---------|
| CapsNet+Q | $9.88\pm_{0.06}$ | $8.49\pm_{1.50}$ | $4.55\pm_{1.13}$ |
| CapsNets+REM | $9.40\pm_{0.55}$ | $6.15\pm_{2.32}$ | $3.85\pm_{0.54}$ |

Table 9: Entropies for models trained with the decoder on Fashion-MNIST and CIFAR10 (test set).

| Model | Familiar | | Novel | | Sparsity |
|-------|----------|---------|-------|---------|----------|
| | Accuracy | Entropy | Accuracy | Entropy | |
| CapsNet+Q ($\phi$) | $90.51\pm_{0.002}$ | $6.25\pm_{1.15}$ | $77.40\pm_{0.006}$ | $5.01\pm_{1.45}$ | 0 |
| CapsNet+REM ($\phi$) | $90.38\pm_{0.005}$ | $3.35\pm_{1.18}$ | $76.98\pm_{0.004}$ | $2.47\pm_{0.96}$ | 50.13 |
| CapsNet+Q ($\psi$) | $87.44\pm_{0.005}$ | $5.42\pm_{1.34}$ | $72.29\pm_{0.005}$ | $5.02\pm_{1.07}$ | 0 |
| CapsNet+REM ($\psi$) | $86.78\pm_{0.005}$ | $3.07\pm_{1.33}$ | $71.89\pm_{0.006}$ | $2.75\pm_{1.44}$ | 60.81 |
| $\gamma$-CapsNet+Q ($\phi$) | $89.62\pm_{0.005}$ | $1.78\pm_{0.82}$ | $75.54\pm_{0.005}$ | $2.72\pm_{1.08}$ | 0 |
| $\gamma$-CapsNet+REM ($\phi$) | $88.43\pm_{0.132}$ | $1.24\pm_{0.61}$ | $74.40\pm_{0.062}$ | $2.05\pm_{0.80}$ | 50.30 |
| $\gamma$-CapsNet+Q ($\psi$) | $85.98\pm_{0.004}$ | $1.87\pm_{0.58}$ | $71.33\pm_{0.012}$ | $2.55\pm_{0.89}$ | 0 |
| $\gamma$-CapsNet+REM ($\psi$) | $85.26\pm_{0.024}$ | $1.52\pm_{0.42}$ | $71.12\pm_{0.025}$ | $2.17\pm_{0.61}$ | 49.61 |
| DeepCaps+Q ($\phi$) | $95.01\pm_{0.005}$ | $7.32\pm_{1.18}$ | $83.18\pm_{0.016}$ | $7.28\pm_{1.66}$ | 0 |
| DeepCaps+REM ($\phi$) | $94.62\pm_{0.005}$ | $6.75\pm_{1.41}$ | $82.49\pm_{0.015}$ | $6.12\pm_{1.91}$ | 34.45 |
| DeepCaps+Q ($\psi$) | $90.16\pm_{0.002}$ | $6.53\pm_{1.77}$ | $79.36\pm_{0.007}$ | $5.14\pm_{1.45}$ | 0 |
| DeepCaps+REM ($\psi$) | $90.13\pm_{0.001}$ | $5.55\pm_{1.49}$ | $78.66\pm_{0.012}$ | $3.92\pm_{1.36}$ | 36.06 |

Table 10: Accuracy (%) and entropy values on the smallNORB test set on familiar and novel viewpoints (elevations $\phi$ and azimuths $\psi$) for CapsNets, $\gamma$-CapsNets and DeepCaps when quantization and REM are applied.

| Model | Familiar | Novel | Sparsity |
|---|---|---|---|
| Efficient-CapsNet ($\phi$) | $90.55\pm_{0.051}$ | $80.68\pm_{0.009}$ | 0 |
| Efficient-CapsNet ($\phi$) | $90.67\pm_{0.113}$ | $80.49\pm_{0.007}$ | 55.34 |
| Efficient-CapsNet ($\psi$) | $90.16\pm_{0.072}$ | $80.34\pm_{0.013}$ | 0 |
| Efficient-CapsNet ($\psi$) | $89.87\pm_{0.044}$ | $79.94\pm_{0.014}$ | 47.91 |
| CapsNet ($\phi$) | $90.62\pm_{0.002}$ | $77.51\pm_{0.004}$ | 0 |
| CapsNet ($\phi$) | $90.51\pm_{0.004}$ | $77.03\pm_{0.003}$ | 50.13 |
| CapsNet ($\psi$) | $87.90\pm_{0.005}$ | $72.37\pm_{0.004}$ | 0 |
| CapsNet ($\psi$) | $86.81\pm_{0.006}$ | $71.99\pm_{0.006}$ | 60.81 |
| $\gamma$-CapsNet ($\phi$) | $90.15\pm_{0.003}$ | $75.89\pm_{0.005}$ | 0 |
| $\gamma$-CapsNet ($\phi$) | $89.92\pm_{0.009}$ | $74.96\pm_{0.007}$ | 50.30 |
| $\gamma$-CapsNet ($\psi$) | $86.11\pm_{0.006}$ | $72.55\pm_{0.008}$ | 0 |
| $\gamma$-CapsNet ($\psi$) | $85.35\pm_{0.016}$ | $71.35\pm_{0.013}$ | 49.61 |
| DeepCaps ($\phi$) | $95.32\pm_{0.004}$ | $83.13\pm_{0.009}$ | 0 |
| DeepCaps ($\phi$) | $94.48\pm_{0.003}$ | $82.42\pm_{0.15}$ | 34.45 |
| DeepCaps ($\psi$) | $91.11\pm_{0.002}$ | $79.53\pm_{0.007}$ | 0 |
| DeepCaps ($\psi$) | $90.15\pm_{0.009}$ | $78.83\pm_{0.011}$ | 36.06 |

Table 11: Accuracy (%) on the smallNORB test set on familiar and novel viewpoints (elevations $\phi$ and azimuths $\psi$) for CapsNets, $\gamma$-CapsNets, DeepCaps and Efficient-CapsNets without quantization.

