# OpenReview forum: "REM: Routing Entropy Minimization for Capsule Networks"
_ICLR.cc/2023/Conference — Submitted to ICLR 2023_

### Official Review · Reviewer_uZJ5 · 2022-10-21

**Confidence:** 4
**Clarity, Quality, Novelty And Reproducibility:** The paper is clearly written and easy…
**Correctness:** 4
**Technical Novelty And Significance:** 2
**Empirical Novelty And Significance:** 2
**Recommendation:** 3

**Strength And Weaknesses:**

Strength:
1) The paper gives detailed instructions on how to extract the parse tree of capsule networks (capsule connections) and how to penalize the sparsity of the connections.
2) The proposed method is easy to be plugged into the current capsule networks, which effectively improves the sparsity of the connections while keeping comparable performance.

Weakness:
1) Lack of novelty. The main contribution of the paper is to prune the capsule connections by penalizing the activations and votes of capsules. While the method for pruning is borrowed from other papers (LOBSTER, Tartaglione et al, 2022).
2) In table 1, the paper combines REM with the current methods and evaluates them on different datasets to show the generalization ability, which is not convincing. To validate the generalization, there should be cross-data evaluation, where the new capsule network should be trained on one dataset and tested on another dataset. As I can see, the results in Table 3 show that the performance is reduced when applying the capsule network to novel viewpoints.
4) The interpretability of the sparse capsules need to be further illustrated, i.e., show the sparse part-whole relations learned by the satisfied capsule networks.

Some details:
(1) Following the formatting instructions of ICLR2023, the abstract must be limited to one paragraph.


**Summary Of The Paper:**

The paper proposes a new routing strategy, Routing Entropy Minimization (REM), for capsule networks. The core idea of REM is to minimize the entropy of capsule parse trees by pruning so that the activated connections between capsules are stronger and fewer, which will improve the interpretability of the capsule networks. Experiments on Fashion-MNIST, SVHN, smallNORB, and CIFAR10 validate that the proposed method achieves comparable performance with a significant lower number of parse trees.

**Summary Of The Review:**

The paper gives a detailed analysis of how to sparsify the connections of capsules. However, the effectiveness and the interpretability of the sparsified networks have not been well discussed and explored. Thus, my recommendation is to reject it in its current form.

---

> ### Author Response · Authors · 2022-11-18
> **Response to Reviewer uZJ5**
>
> We thank the Reviewer for raising this important concern. Please see our detailed response below, where we addressed all your raised points. We highlighted in blue the modifications in the paper with respect to the first submission.
>
> **W1: Contribution of the paper**
>
> Our main contribution is not only to prune the capsule connections by penalizing the activations and votes of capsules, but also to minize the entropy of the routing algorithm coefficients to improve its interpretability, as shown in Tables 1, 2 and 3. In CapsNets literature there is no demonstration that these networks explicitly model interpretable nodes/concepts which are hierarchically organized in capsule networks. Furthermore, we also show in Section 3.3 how we can achieve low entropy settings by pruning CapsNets during training.
> We also show qualitative results in Figures 4 and 8 exploiting the parse tree of the predicted class capsule as a saliency map, which is the most frequently used method of interpretation in deep learning. In our work, we extract the saliency maps during the forward pass of the network as described in Section A.1.5 (Figure 6), without requiring the backward pass. Therefore, our contribution relies also on the fact that we visualize what a CapsNet really learned, since many works in the literature (for example the ones already cited in the paper) focus mostly on improving the performances of CapsNets without showing the parts-wholes hierarchies extracted by a CapsNet.
>
> **W2 (a): Cross-data evaluation**
>
> We agree with the Reviewer that training a model on k-1 folds of the training set and testing it on the remaining part of the data is a better way to validate the generalization. But in CapsNets literature, for example the CapsNets model introduced in [1,2,3,4] and that we used in our paper for validating our technique REM, cross-validation is not used to test the generalization ability of these models. This is the reason why in our work we did not employ cross-validation either.
>
> **W2 (b): Results for novel viewpoints**
>
> As regards the results in Table 3, the performances in the column "Novel" are lower than the ones in the column "Familiar" because in the former setting we tested the networks on viewpoints never seen during training. In this specific setting, every model that we are aware of perform worse than on familiar viewpoints. To the best of our knowledge, there is no CapsNet model that achieves the same performances on novel and familiar viewpoints. It would be an ideal situation. Therefore, since we agree that this can be misleading, we added Table 11 in the revised Appendix where the performances of the CapsNets architectures are shown without applying REM. This shows that the performance is reduced when applying CapsNets to novel viewpoints even when the pruning and quantization steps are not applied. The important thing for our work is that, when applying REM, we achieve similar robustness results as the original architecture but with lower entropy.
>
>
> **W3: Interpretability of the sparse capsules**
>
> Thank you, we agree this is important, and indeed we have improved the description of Figure 4.
> As already mentioned in the answer to weakness 1 (W1), in Figures 4 and 8 we actually show the part-whole relations learned by a CapsNet. In Section A.1.5 (Figure 6) we describe the pipeline to illustrate these relationships. Since CapsNets are typically stacked on top of convolutional layers, capsules in these layers also have a spatial connotation. Therefore, we exploit capsule activations and the coupling coefficients corresponding to the predicted class to extract a 2D saliency map, where each pixel value corresponds to the probability that an active low-level capsule (a part) is connected to the higher-level capsule (the whole, in this case the higher-level capsule is the object class capsule). Note that this pipeline does not require any backward pass, to compute these saliency maps we just need a forward pass of an already trained CapsNet model for a given input. For example, in Figure 4, we can see that the part-whole hierarchies extracted with REM are more succinct and human-interpretable. So if we ask why the network classifies that object as a horse, thanks to these saliency maps, we can understand that it is because that object has a head, main and legs. Or why does the network see an airplane in the input image? Because it detects the wings and tail as discriminative parts of the airplane.
>
>
> **Additional comments**
>
> We fixed the abstract following the formatting instructions of ICLR2023.

---

> > ### Author Response · Authors · 2022-11-18
> > **References**
> >
> > **References**
> >
> > [1] Sara Sabour, Nicholas Frosst, and Geoffrey E Hinton. Dynamic routing between capsules. (NeurIPS 2017)
> >
> > [2] Jathushan Rajasegaran, Vinoj Jayasundara, Sandaru Jayasekara, Hirunima Jayasekara,
> > Suranga Seneviratne, and Ranga Rodrigo. DeepCaps: Going Deeper with Capsule Networks (CVPR 2019)
> >
> > [3] Vittorio Mazzia, Francesco Salvetti and Marcello Chiaberge. Efficient-CapsNet: capsule network with self-attention routing. (Nature Scientific reports 2021)
> >
> > [4] David Peer, Sebastian Stabingerm, and Antonio Rodriguez-Sanchez. Increasing the adversarial robustness and explainability of capsule networks with γ-capsules

---

> > ### Comment · Reviewer_uZJ5 · 2022-11-25
> > **Comments after response**
> >
> > 1. Contribution
> > From my perspective, minimizing the entropy is to penalize the activations and votes of the capsules, so that the connections between the capsules are sparse. As the author claims, minimizing entropy can improve the interpretability of networks. Table 1, 2, 3 are used to validate that the proposed method can effectively sparse the connections but not the interpretability. Note that the sparse for interpretability is the motivation for this paper, it needs to be further validated. Besides, the saliency map is not the only way to show the improvement of interpretability. For example, the CapsNet shows that each dimension of the capsule presents a different attribute of the digits. Is the CapsNet+REM shows better decomposition results? More experiments need to be designed and explored by the authors.
> >
> > 4. Interpretability
> > The authors are suggested to show the visualization results to show the interpretability improvement of different methods, similar to Table 1.

---

> > > ### Author Response · Authors · 2022-11-26
> > > **Response to Reviewer uZJ5**
> > >
> > > We thank the Reviewer for these helpful comments.
> > >
> > > **Beyond saliency maps**
> > >
> > > We agree that exploring additional methods apart from saliency maps is a good point for showing the improvement of interpretability. As regards our work, we focus not on the digit capsule representations but on the connections between capsules, this is why we exploit saliency maps built upon the coupling coefficients to show better interpretability. But having said that, as regards last layer capsule vectors, as explained in Section A.3.1, [1] report that, when a perturbed digit activity vector is fed to the decoder, such perturbation affects the reconstruction, showing that these capsules are approximately equivariant. Actually, we conducted these experiments also with CapsNets+REM (Figure 9). We can see that CapsNets+REM with low entropy are still approximately equivariant to many transformations. Regarding better decomposition results with this method, it is hard for us to state that CapsNets+REM are better in this aspect. Therefore, if the Reviewer has other ideas to show these decomposition properties apart from feeding a perturbated vector to the decoder, we are pleased to follow other suggestions to improve our paper.
> > >
> > > **Additional visualizations**
> > >
> > > We will add saliency maps visualization with the other CapsNets models reported in Table 1.
> > >
> > > **References**
> > >
> > > [1] Sara Sabour, Nicholas Frosst, and Geoffrey E Hinton. Dynamic routing between capsules. (NeurIPS 2017)

---

### Official Review · Reviewer_1ErG · 2022-10-24

**Confidence:** 1
**Correctness:** 3
**Technical Novelty And Significance:** 3
**Empirical Novelty And Significance:** 3
**Recommendation:** 6

**Clarity, Quality, Novelty And Reproducibility:**

The paper appears to be technically sound, and the experiments are relatively sufficient, which supports the initial claims.


**Strength And Weaknesses:**

Strengths:
1 The motivation of this paper is clearly described.
2 Empirical results indicate that the proposed method achieves good performance.
3 The paper is well organized.

Weaknesses:
1 Pruning and Quantization are standard methods in compression and acceleration literature. Intuitively, the paper just applies these two techniques to the capsule networks. What are the contributions of these techniques?


**Summary Of The Paper:**

RSM is proposed to minimize the entropy of the parse tree-like structure in Capsule Networks. Pruning is used to reduce the entropy of the model parameters distribution. These methods generate a significantly lower number of parse trees, with no performance loss. Experimental results show the effectiveness of the proposed method.


**Summary Of The Review:**

 I am not familiar with this topic. I score this paper marginally above the acceptance threshold, based on the solid contribution by minimizing the entropy of the connections and comprehensive experiments.

---

> ### Author Response · Authors · 2022-11-18
> **Response to Reviewer 1ErG**
>
> Thanks for your interest and feedback on our work.
>
> In CapsNets literature there is no demonstration that these networks explicitly model interpretable nodes/concepts, which are hierarchically organized in multi-layered capsule networks.
> The motivation of our entropy based approach REM is depicted in Figure 2. We want to lower the entropy in order to extract fewer and stronger parse trees, achieving high generalization, so that understanding the relationships between capsules is more straightforward.
> REM pipeline employs an iterative pruning strategy and quantization stage to CapsNets as shown in Figure 1. The pruning method LOBSTER [1] is applied during training, namely at each epoch we prune the CapsNet model and then we retrain the network. Therefore, we introduce a sparsity constraint during training, which is crucial to lower the entropy, as described in Section 3.2. We apply the quantization step during inference in order to compute the number of parse trees extracted from the model on a given dataset.
> In Section 4.1 we analyze in-depth the benefits of pruning and quantization steps separately and in Table 3 we show when we apply all this steps together the entropy measure for CapsNets+REM is lower compared to CapsNets+Q, namely, REM has successfully extracted a lower number of stronger parse trees on different datasets.
> We also show qualitative results in Figures 4 and 8 exploiting the parse tree of the predicted class capsule as a saliency map, which is most frequently used method of interpretation in deep learning.
>
> **References**
>
> [1] Enzo Tartaglione, Andrea Bragagnolo, Attilio Fiandrotti, and Marco Grangetto. Loss-based sensitivity regularization: Towards deep sparse neural networks. (Neural Networks 2022)

---

### Official Review · Reviewer_Ni7P · 2022-10-24

**Confidence:** 4
**Correctness:** 4
**Technical Novelty And Significance:** 2
**Empirical Novelty And Significance:** 3
**Recommendation:** 6

**Clarity, Quality, Novelty And Reproducibility:**

The paper is well presented and includes useful experiments - I think experiments need to be contextualised with respect to other work, irrespective of whether the method achieves SOTA or not. Reproducibility is not the strongest part of this paper (no code is included), but they provide enough information to reproduce some parts of this work.


**Strength And Weaknesses:**

Strengths: a) There is some new methodology described to extract parse trees and use pruning and quantization in capsule networks b) Results are promising and it seems that the method does work competitively with respect to other solutions presented in the literature. However, I do think authors are missing two important works in the area, i.e. subspace capsules (Edraki et al.) and Capsule Routing via Variational Bayes (De Sousa Ribeiro et al.)

Weaknesses: a) Table 3 and other tables (wherever relevant) needs to include other methods as well (e.g. ones I mentioned above and also ones already cited). Authors do not need to reimplement anything as all these (or most of them) methods do have results in such datasets.

The abstract starts: "Capsule Networks ambition" this needs to be paraphrased, as it is our ambition to make them interpretable not their ambition - or "for Capsule Networks to be interpretable we need to.....



**Summary Of The Paper:**

I would have hoped to be clearer on how all the components proposed fit together, but in a nutshell, the authors propose a way to prune CapsNets without compromising performance, and they do so by pushing the model parameters to low entropy spaces. They achieved competitive results in some datasets they compared with, demonstrating that one can reduce entropy and cardinality in the parse trees extracted.

**Summary Of The Review:**

This is a good submission which helps to provide a new dimension to capsule neural networks. There is a good experimental setup presented but I think it requires further improvements to demonstrate correspondence with other methods presented in the literature - this is the major limitation of this paper, as it is hard to put this paper in context (e.g. to compare with other methods which include number of parameters etc.)

---

> ### Author Response · Authors · 2022-11-18
> **Response to Reviewer Ni7P**
>
> Thanks for the insightful and constructive feedback. Please see our detailed response below. We highlighted in blue the modifications with respect to the first submission.
>
> **W1: Subspace capsules (Edraki et al.) and Capsule Routing via Variational Bayes (De Sousa Ribeiro et al.)**
>
> We agree with the Reviewer that subspace capsule networks (SCNs) [1] and variational bayes capsule networks (VBCaps) [2] are two important works in the capsule area, and they have been included in the state-of-the-art section. Unfortunately, we can not include these works in our experimental analysis for the reasons detailed in the following.
> Regarding SCNs, there is no routing, therefore they do not model the part-whole relationships between entities through the coupling coefficients that form the parse trees. So there is no point in applying our REM technique, which relies on the entropy of the parse trees, to this special capsule setting. Furthermore, they are applied in a GAN framework, while all the other models that we employed in our work are trained in a supervised classification setting.
> Concerning VBCaps, they propose a new capsule routing algorithm derived from Variational Bayes for ﬁtting a mixture of transforming gaussians [2], inspired by the expectation-maximization routing algorithm. We tried both networks implemented in https://github.com/fabio-deep/Variational-Capsule-Routing/blob/master/src/capsnet.py but we noticed that the coupling coefficients computed by these models require high numerical precision. In fact, when we tried to quantize the coupling coefficients of the routing algorithm without pruning, the performance dropped by ~60% in accuracy (K=11 quantization levels on MNIST) We conjecture that this can be due to the fact that in VBCaps the routing mechanism is not made independent enough from the feature extraction phase. Indeed, VBCaps feature extractor network, before stacking capsule layers, is composed of only one convolutional layer with 64 5x5 kernels. We also tested VBCaps using the same convolutional layer with 256 9x9 kernels but, also in this higher dimensional setting, we noticed a major drop in performance when the quantization phase is applied.
>
> **W2: Additional results**
>
> We agree that Table 3, where we tested our technique to novel viewpoints on smallNORB dataset with Efficient-Caps [3], needs to include other methods already cited in our work. Therefore, we added Table 10 in the Appendix (revised paper) that includes results also for CapsNets [4], γ-CapsNets [5] and DeepCaps [6] on smallNORB. We can see that our technique is robust to novel viewpoints even when applied to these models.
>
> **W3: Abstract**
>
> We agree that the abstract should be paraphrased since it was misleading. We changed "Capsule Networks aim to build an interpretable and biologically-inspired neural network model" to "Capsule Networks are biologically-inspired neural network models, but their interpretability still needs to be further investigated".
>
>
> **Additional comments**
>
> Please notice that source code is included in the supplementary zip material.
>
> **References**
>
> [1] Marzieh Edraki, Nazanin Rahnavard, and Mubarak Shah. SubSpace Capsule Network (AAAI 2020)
>
> [2] Fabio De Sousa Ribeiro, Georgios Leontidis, and Stefanos Kollias. Capsule Routing via Variational Bayes (AAAI 2019)
>
> [3] Vittorio Mazzia, Francesco Salvetti and Marcello Chiaberge. Efficient-CapsNet: capsule network with self-attention routing. (Nature Scientific reports 2021)
>
> [4] Sara Sabour, Nicholas Frosst, and Geoffrey E Hinton. Dynamic routing between capsules. (NeurIPS 2017)
>
> [5] David Peer, Sebastian Stabingerm, and Antonio Rodriguez-Sanchez. Increasing the adversarial robustness and explainability of capsule networks with γ-capsules
>
> [6] Jathushan Rajasegaran, Vinoj Jayasundara, Sandaru Jayasekara, Hirunima Jayasekara, Suranga Seneviratne, and Ranga Rodrigo. DeepCaps: Going Deeper with Capsule Networks (CVPR 2019)

---

### Official Review · Reviewer_LZej · 2022-10-25

**Confidence:** 4
**Correctness:** 3
**Technical Novelty And Significance:** 3
**Empirical Novelty And Significance:** 3
**Recommendation:** 5

**Clarity, Quality, Novelty And Reproducibility:**

The quality is good. And the idea to reduce the entropy of routing is novel. But the paper is of low clarity in terms of motivation and method description, thus leading to reproducibility.

**Strength And Weaknesses:**

## Strength
- This work studies the interpretability of dynamic routing algorithms, which is very interesting to the community.
- The paper reports lots of experimental analysis and reveals some interesting results  -- e.g., entropy changes curves along the training.

## Weaknesses
The paper has several weaknesses which I'll detail below:

Writing:
- Motivation for the method is not strong. Fewer parameters are better for computation overhead but it doesn't mean that we can better interpret the capsule network. However, this paper motivates their story from the viewpoint of interpretability, which I personally don't agree. I didn't see how the routing of lower entropy can help understand the behavior of the network. I would recommend having better motivation.
- Low clarity in the method description. I sometimes have difficulties understanding the idea/motivation behind the design choices. For example, in Eq. 16, why is the method trained with a special optimizer? Any reason behind this? I would recommend a better justification for this choice. And I'm also not very sure how you define entropy.


Method:
- The paper claims the proposed method minimizes the entropy of the routing algorithm. It seems to be an over-claim from my perspective. Because I didn't see any regularizers/optimization loop inside. I don't quite get how entropy is minimized for dynamic routing. Is that just through a connection pruning step?

Experimental results:
- I didn't find any experiments that support the proposed method. By that I mean the results show the benefit/advantages of the proposed algorithm compared with other methods in terms of efficiency. The only comparison seems to be an ablation study of the proposed method. It's not solid.


**Summary Of The Paper:**

This paper presents a method (REM) to reduce the connection in the dynamic routing step of the capsule network, thereby reducing the entropy of routing. By doing so, the paper shows improved interpretability of the capsule networks. Specifically, the work proposes to simply prune connection via quantization of coupling coefficients in routing and thus get the parse trees. The paper validates its method and analyzes the routing in multiple datasets.

**Summary Of The Review:**

The paper proposes an interesting idea for training the capsule network while pruning the connection of routing. The experimental results show some improvement. But it's very marginal. And also I'm not sure about the motivation behind it. In other words, I'm not very clear about its contribution. Furthermore, I feel the experimental validation is not solid, as I detailed above. I thus tend to reject this paper. But I'm very happy if the author can provide feedback just in case I misunderstand anything.

---

> ### Author Response · Authors · 2022-11-18
> **Response to Reviewer LZej**
>
> We thank the Reviewer for raising this important concern. Please see our detailed response below.
>
> **W1: How low entropy routing helps CapsNets interpretability**
>
> The aim of our work is to provide a better interpretation of the routing mechanism, which computes the coupling coefficients between low level and high level capsules. The routing mechanism is extremely important in making CapsNets an interpretable architecture [1]: our goal is to dig more, understanding how features are manipulated at such level. These relationships model the hierarchy of parts and wholes of an object, carving-out a parse tree. As mentioned in Section 4.1, the problem of CapsNets is that these relationships do not emerge really well. In fact there are too many active co-coupled capsules, namely their entropy is very high, making the routing algorithm connections still difficult to understand. In an ideal situation, to extract a proper scene graph or parse tree from the input image, a lower level capsule should be connected with probability 1 to only one parent capsule, with probability 0 to all the other high level capsules.
> The motivation of our entropy based approach REM is depicted in Figure 2. We want to lower the entropy in order to extract fewer and stronger parse trees, achieving high generalization, so that understanding the relationships between capsules is more straightforward. In order to achieve this property, we show in Figure 1 (in the revised paper ) the overall pipeline of our technique. In more details, we employ:
>  - the pruning method LOBSTER [2] during training, namely at each epoch we prune the CapsNet model and then we retrain the network, in order to lower the entropy of the parse tree extracted as described in Section 3.2.
>  - the quantization step during inference in order to compute the dictionary of the parse trees extracted from the model on a given dataset as described in Section 3.1.
> In Section 4.1 we analyze in-depth the benefits of pruning and quantization steps separately and in Table 3 we show when we apply all this steps together the entropy measure for CapsNets+REM is lower compared to CapsNets+Q, namely, REM has successfully extracted a lower number of stronger parse trees on different datasets.
>
> We also show qualitative results in Figures 4 and 8 exploiting the parse tree of the predicted class capsule as a saliency map. Typically, methods such as Grad-CAM [3] are applied to extract the saliency maps. In our work, we extract the saliency maps during the forward pass of the network as described in Section A.1.5 (Figure 6), without requiring the backward pass. Therefore, our contribution relies also on the fact that we visualize what a CapsNet really learnt, since many works in the literature (for example the ones already cited in the paper) focus mostly on improving the performances of CapsNets without showing the parts-wholes hierarchies extracted by a CapsNet.
>
> **W2: Why is the method trained with a special optimizer? (Eq. 16)**
>
> Our method is not trained with a special optimizer. Eq. 16 is the update rule of [2], which penalizes the parameters by their gradient-weighted L2 norm. As stated in Section 3.3, this update rule can be used with any optimizer such as SGD, SGDM, Adam etc. In our experiments we used Adam since for CapsNets we achieved better convergence.
>
> **W3: How do you define entropy.?**
>
> The entropy for a single object class is defined in Eq. 8. The total entropy is simply the average of the entropy of all object classes (Eq. 9). Intuitively, we consider each parse tree as a string and we compute the entropy according to their occurrences in the input dataset.
>
> **W4: I don't quite get how entropy is minimized for dynamic routing. Is that just through a connection pruning step?**
>
> In Section 3.3 we show how we can implicitly minimize routing entropy by forcing a sparse and organized structure in
> the coupling coefficients, that is achieved by enforcing sparsity in the W_ij learnable transformation matrices. In order to obtain sparse W_ij, we use pruning [2] as a proxy. Therefore, the entropy of the parse trees is minimized during training since we prune CapsNets during training.
>
> **W5: Benefit/advantages of the proposed algorithm compared with other methods in terms of efficiency.**
>
> As stated in Section 1, our work we do not focus on the efficiency of CapsNets, we show the number of pruned parameters only for the sake of completeness. This paper aims to improve interpretability of the routing algorithm in
> CapsNets. We employ a pruning strategy not for efficiency but to drive the model parameters distribution towards low entropy configurations.

---

> > ### Author Response · Authors · 2022-11-18
> > **References**
> >
> > [1] Sara Sabour, Nicholas Frosst, and Geoffrey E Hinton. Dynamic routing between capsules. (NeurIPS 2017)
> >
> > [2] Enzo Tartaglione, Andrea Bragagnolo, Attilio Fiandrotti, and Marco Grangetto. Loss-based sensitivity regularization: Towards deep sparse neural networks. (Neural Networks 2022)
> >
> > [3] Ramprasaath R. Selvaraju, Michael Cogswell, Abhishek Das, Ramakrishna Vedantam, Devi Parikh, and Dhruv Batra. Grad-cam: Visual explanations from deep networks via gradient-based localization. (ICCV 2017)

---

### Decision · Program_Chairs · 2023-01-20

**Decision:**

Reject

**Justification For Why Not Higher Score:**

Two reviewers argue that experimental results are insufficiently convincing and more work is required to analyze the claims of interpretability.  The AC concurs with these reviewers.

**Justification For Why Not Lower Score:**

N/A

**Metareview: Summary, Strengths And Weaknesses:**

The paper presents a method for pruning capsule networks, with an aim of improving model interpretability.

Reviewer 1ErG indicates low confidence in ability to access the paper.  Reviewer Ni7P, though rating marginal accept, comments on experimental weaknesses in terms of comparison to other methods.  Reviewers uZJ5 and LZej similarly voice concerns over insufficient experimental validation, with Reviewer LZej stating, "I didn't find any experiments that support the proposed method."  Reviewer uZJ5 raises additional concerns over novelty.

While the author response adds an additional table of experiments (Table 10 in Appendix), these results appear far from convincing as accuracy differences of the proposed method over baselines are both marginal and sometimes negative (in exchange for sparsity).  Reviewer uZJ5's reply in the discussion thread indicates further unresolved questions over interpretability and a request for corresponding additional experiments.

The Area Chair agrees with Reviewer uZJ5 and LZej's overall assessments of the paper.  The experimental case appears insufficiently developed and additional work is needed to establish the claims of interpretability.